



# Performance analysis of the NanoScan SMPS and the Mini WRAS Ultrafine Aerosol Particle Size Spectrometers

Ajit Ahlawat[1], Kay Weinhold[1], Jesus Marval[2], Paolo Tronville[2], Ari Leskinen[3,4], Mika Komppula[3], Holger Gerwig[5], Lars Gerling[6], Stephan Weber[6], Rikke Bramming Jørgensen[7], Thomas Nørregaard Jensen[8], Marouane Merizak[9], Ulrich Vogt[9], Carla Ribalta[10], Mar Viana[11], Andre Schmitz[12], Maria Chiesa[13], Giacomo Gerosa[13], Lothar Keck[14], Markus Pesch[14], Gerhard Steiner[14], Thomas Krinke[15], Torsten Tritscher[15], Wolfram Birmili[16], Alfred Wiedensohler[1]

[1]Leibniz Institute for Tropospheric Research, Permoserstrasse 15, 04318 Leipzig, Germany

[2]Politecnico di Torino - DENERG, Corso Duca degli Abruzzi 24, 10129 Turin, Italy

[3]Finnish Meteorological Institute, Yliopistonranta 1 F, 70210 Kuopio, Finland

[4]University of Eastern Finland, Yliopistonranta 1 F, 70210 Kuopio, Finland

[5]Umweltbundesamt, Paul-Ehrlich-Straße 29, 63225 Langen, Germany

[6]Technische Universität Braunschweig, Institute of Geoecology, Langer Kamp 19c, 38106 Braunschweig, Germany

[7] Norwegian University of Science and Technology, Department of Industrial Economics and Technology Management, 7491 Trondheim, Norway

[8]Center for Air and Sensor Technology, Danish Technological Institute, DK-8000 Aarhus C, Denmark

[9]Institut für Feuerungs- und Kraftwerkstechnik (IFK), Universität Stuttgart, Pfaffenwaldring 23 70569 Stuttgart, Germany

[10]Det Nationale Forskningscenter for Arbejdsmiljø, Lersø Parkallé 105, 2100 København Ø, Denmark

[11]Institute of Environmental Assessment and Water Research (IDAEA-CSIC), Jordi Girona 18, 08034 Barcelona, Spain

[12]Wessling GmbH, Oststraße 7, 48341 Altenberge, Germany

[13]Università Cattolica del Sacro Cuore, Department of Mathematics and Physics, Via della Garzetta 48, 25133 Brescia, Italy

[14]GRIMM Aerosol Technik Ainring GmbH & Co. KG, Dorfstrasse 9, 83404 Ainring, Germany

[15]TSI GmbH, Neuköllner Str. 4, 52068 Aachen, Germany

[16]Umweltbundesamt, Corrensplatz 1, 14195 Berlin, Germany

*Correspondence to*: Ajit Ahlawat (ahlawat@tropos.de), Alfred Wiedensohler (ali@tropos.de)



**Abstract**

In aerosol science, there is an increasing interest to perform mobile measurements to obtain number size distribution of ultrafine particles (UFP), using portable instruments based on unipolar charging and size segregation by electrical particle mobility. Applications of such measurements range from ambient and indoor aerosol studies to source identification in work environments. However, knowledge on the actual measurement uncertainties of these portable instruments under various conditions has been limited. This investigation presents results from an intercomparison workshop conducted at the World Calibration Center for Aerosol Physics (WCCAP) in Leipzig, Germany, in January 2020. Manufacturers and users were invited to have their portable instruments tested and compared against reference instrumentation for particle number size distributions (PNSD) and total particle number concentration (PNC). In particular, the performances and uncertainties of the NanoScan SMPS (Scanning Mobility Particle Sizer) Model 3910 (TSI Inc.) and the Mini Wide Range Aerosol Spectrometer (WRAS) Model 1371 (Grimm Aerosol Technik) were investigated extensively against the WCCAP Mobility Particle Size Spectrometers (MPSS) and Condensation Particle Counters (CPC). A total of 11 TSI NanoScan SMPS and 4 GRIMM Mini WRAS instruments were characterized for ambient aerosols as well as lab-generated aerosols.

The workshop results affirm that the portable instruments must be serviced and calibrated annually or prior field studies to provide measurements within the given uncertainties. It should be noted that users should carry out timely service, maintenance and calibration of portable instruments at their facilities. During initial inspection, non-serviced NanoScan SMPS instruments overestimated a dominant ultrafine aerosol mode by 120% at around 80 nm. Maintenance and servicing improved the performance. Overall, the performance of NanoScan SMPS instruments improved for the ultrafine aerosol mode while the PNC in the fine aerosol mode still overestimated by up to 80%. The latter effect seems to be systematically related to the unipolar charging of particles, and the reduced sensitivity of electrical particle mobility with increasing particle size above 200 nm. Due to shift in the second mode of bimodal distribution, particles are overcounted around 100 nm. With regard to the integral PNC, some of the NanoScan SMPS found to be in good agreement (i.e. within 20%) compared to the reference CPC. In addition, a reasonably good unit-to-unit agreement within ±20% was found for NanoScan SMPS instruments. The Mini WRAS instruments, after proper cleaning and servicing, provided improved results within ±15% deviation in PNC in the ultrafine aerosol mode. Overall, most of the GRIMM Mini WRAS instruments (operating with software version 10.0) agrees well with PNC (i.e. 10-50%) when the ultrafine mode was dominant. Conversely, PNC of the fine aerosol mode was systematically underestimated by 60% above 100 nm. Except for one instrument, the integral PNC of the GRIMM Mini WRAS spectrometers were within an uncertainty range of ±20% compared to the reference CPC. Additionally, it is important for users to note that the Mini WRAS performed significantly better when using software version 10.0 compared to version 8.2.

The workshop results suggest that despite the above-mentioned uncertainties, these portable instruments are suited for mobile ultrafine particle measurements to detect relative differences in the PNSD such as source apportionment studies of ultrafine particles at work places or outdoors near sources.

**Introduction:**

Ultrafine aerosol particles (UFP), defined as airborne particles smaller than 100 nm in diameter, have gained increasing attention due to their potential role with regard to human health (Kwon et al., 2020) and climate (Kerminen et al., 2012). UFP are inadvertently emitted into the atmosphere by a number of processes, with combustion sources such as combustion engines, stationary power generation, and natural forest fires counting among the most significant (Lighty et al., 2000). Other sources of UFP include atmospheric nucleation as a result of photochemical processes (Kulmala et al., 2014), and even abrasive processes such as break wear (Jansson et al., 2010). UFP have significant progression rates with respect to aerosol dynamic processes such as coagulation and deposition. Considering the time-dependency of source emission profiles, the spatial and temporal variations of UFP concentrations in the atmosphere may be large (Ning and Sioutas, 2010; von Bismarck-Osten et al., 2013; Kumar et al., 2014).

A main hypothesis for their adverse health effects is their small size, allowing UFP to penetrate deep into the alveolar region of the human lung (Kwon et al., 2020), cause size-dependent inflammatory effects (Brown et al., 2001), and translocate to other organs such as the brain (Oberdörster et al., 2005). Atmospheric UFP contains significant fractions of refractory combustion particles, which may not readily dissolve upon inhalation but can instead remain in human tissue for long periods. Besides a refractory core of elemental carbon, they include organic coatings with substances





of enhanced toxicity such as PAH (polycyclic aromatic hydrocarbons). Such particle types, in combination with
particle surface area, have been proposed as a surrogate for particle-induced health effects (Schmid and Stoeger, 2016).
A further concern related to UFP is engineered nanoparticles, which overlap with the size range of unintended UFP
(Madl and Pinkerton, 2009). Health effects of environmental pollutants on populations are usually determined by
epidemiological methods. Although having grown over the past two decades, the overall epidemiological evidence on
the health-effects of environmental UFP in humans has remained scarce and contradictory (Ohlwein et al., 2018). This
is due partly to the lack of suitable environmental data sets for UFP.
Owing to their small size, UFPs contribute only little to the quantitative measurement of mass-based metrics ($PM_{10}$,
$PM_{2.5}$ or $PM_1$) or light scattering. This limitation also affects attempts to determine UFP chemical composition.
Instead, sensitive techniques based on physical particle counting have been developed to accurately measure UFP
number concentrations and particle size distributions (Kuhlbusch et al., 2011). Useful metrics for UFP include total
particle number concentration (PNC or TNC), and the particle number size distribution (PNSD). From a PNSD,
particle number and surface area concentrations can be derived for any desired particle diameter interval including the
UFP range.
High quality instrumentation to determine UFP-related parameters include condensation particle counters (CPC) and
the mobility particle size spectrometer (MPSS). A standard MPSS uses a bipolar diffusion charger to bring the aerosol
particle population into a well-known bipolar charge equilibrium (Wiedensohler et al., 1988). In an MPSS, particle
number size distributions are calculated from electrical mobility distributions employing the size-dependent bipolar
charge distribution in an inversion routine (Pfeifer et al., 2014). Due to their high particle size resolution MPSS data
describe the physical properties of a particle population between 0.01 and 1 µm. An intercomparison between
concurrent MPSS and CPC measurements is useful to assure the quality of UFP measurement data by comparing a
size-selective and an integral aerosol measurement. A considerable body of atmospheric measurement data on PNSD
and total PNC data has been collected by various research groups using MPSS and CPC instrumentation. Significant
observations have been made at least since the 1990s, and have been extended to any kind of region of the globe -
remote, continental, urban, roadside, and industrial (Kecorius et al., 2017; Gani et al., 2019; Gong et al., 2020). MPSS
and CPC now form integral part of several continuously operating networks including ACTRIS (Aerosol, Clouds and
Trace gases Research Infrastructure, https://www.actris.eu), GUAN (German Ultrafine Aerosol Network; Birmili et
al., 2016), and multi-center health studies like RUPIOH (Aalto et al., 2005), UFIREG (Lanzinger et al., 2016) and
the 8 European cities study (Stafoggia et al., 2017).
MPSS and CPC instrumentation has also been applied to measure UFP concentrations in workplace environments
(Kuhlbusch et al. 2011; Koivisto et al. 2014; Fonseca et al. 2015a, b; López et al., 2022). Indoor UFP concentrations
using a MPSS have, however, remained scarce in comparison (Zhao et al., 2020), and we are not aware of any
continuous observations indoors. In summary, there is a growing need to measure PNSD and PNC in various locations
and under different conditions (e.g., Wehner et al. 2002; Costabile et al. 2009; Asmi et al. 2011; Cusack et al. 2013).
In addition, there were some intercomparison experiments reported between stationary MPSSs and CPCs (Asbach et
al. 2012; Wiedensohler et al. 2012; Kaminski et al. 2013; Price et al. 2014). While stationary MPSS or CPC
instruments will be the preferred solution for long-term monitoring and high quality laboratory and field experiments,
the inherent limitations of a standard MPSS with respect to weight, dimension, and power requirement may hamper
their application in mobile settings, or when only quick estimates of a UFP number size distributions are necessary.
The use of a radioactive aerosol bipolar diffusion chargers in a standard MPSS may further hamper its deployment
under the safety standards in many countries.
Consequently, commercial manufacturers have developed more lightweight and portable instruments, which can
complement the radius provided by stationary MPSS instruments. Based on a recent survey of the actual use of these
instruments in the scientific community, two portable instruments were identified for this investigation: The NanoScan
SMPS model 3910 (TSI Inc.) and the Mini WRAS spectrometer 1371 (Grimm Aerosol Technik). The major
advantages of these instruments are easy to use, fast, portable, battery operated, relatively small dimension, and use
of a non-radioactive unipolar charger etc. Additionally, the charging efficiency of unipolar chargers is much higher
than bipolar chargers. A higher time resolution of these portable instruments may also be advantageous for short-term
measurements in environments with a more dynamic aerosol such as exposure assessment in occupational hygiene
settings (Jorgensen et al., 2020). However, some technological choices taken in these mobile instruments imply that
some processes such as charging and mobility classification tend to be less well defined than in a standard MPSS.
This may lead to deviations in the resulting PNSD and PNC in comparison with standard MPSS and CPC instruments,
which will be investigated in this paper.



The TSI NanoScan SMPS model 3910 and the GRIMM Mini WRAS spectrometer 1371 spectrometer use a unipolar
diffusion charger. In contrast to bipolar charging, unipolar charging is associated with additional uncertainties. For
instance, it is known that pre-charged aerosol particles have an impact on the charging efficiency (Qi et al., 2009;
Kaminski et al., 2013). Using a unipolar diffusion charger in conjunction with pre-charged aerosol particles could lead
to a poorly defined unipolar charge distribution. In such cases, the data inversion will not be performed correctly, and
the resulting PNSD will be distorted. Furthermore, unipolar diffusion charging leads to a decreasing sensitivity of the
mean electrical mobility with increasing particle diameter in the fine aerosol mode. Instruments having a unipolar
charge inversion mechanism use an artificial inversion matrix, which cannot compensate for the insensitivity of the
electrical mobility, leading to an overestimation of the PNSD below 200 nm and underestimation above 200 nm. In
practice, this limits the application of such classification devices to the range below 200 nm. It is thus important to
evaluate the performance of the new portable instruments in view of how the aforementioned limitations may actually
be relevant in practice. The most important parameters for a performance evaluation of portable instruments are, a) an
inter-comparison with reference CPC and MPSS, b) checking the unit-to-unit variability, c) flow checks, and d) the
sizing calibration with certified PSL particles (except for instruments with limited size resolution).
So far, intercomparison studies between portable instruments such as TSI NanoScan SMPS model 3910 and the
GRIMM Mini WRAS spectrometer 1371 and stationary MPSS have been limited. Only a few studies were conducted
for the TSI NanoScan SMPS model 3910 instruments (Tritscher et al., 2013; Stabile et al., 2014; Hsiao et al., 2016;
Fonseca et al., 2016). These studies were only limited to either using laboratory-generated test aerosols such as NaCl,
Ag, polystyrene latex, ammonium sulfate $(NH_4)_2SO_4$ particles, di-ethyl hexyl sebacate (DEHS), $TiO_2$, and diesel soot
particles) or using indoor aerosols. Yamada et al. (2015) tested the performance of the TSI NanoScan SMPS model
3910 using nano-$TiO_2$ powder as a test aerosol. They found large differences in PNSD when test aerosols were used
and could not explain the reasons. However, they found that the measured PNSD for indoor aerosols was quite
consistently measured by the TSI NanoScan SMPS model 3910 except for particles greater than 200 nm. Another
recent study comparing portable instruments in exposure environments reports large variations between nanoparticle
measurements and results for the four scenarios (inert metal gas (MIG) welding, polyvinyl chloride (PVC) welding,
cooking, and candle-burning) tested (Jorgensen et al., 2019). Stabile et al., (2014) compared the TSI NanoScan SMPS
model 3910 and a reference SMPS with various polydisperse test aerosols under laboratory conditions. They found
that the agreement was best for spherical particles. Vo et al., (2018) showed a performance comparison of field-
portable instruments (including TSI NanoScan SMPS model 3910) to a reference MPSS challenged by monodisperse
and polydisperse sodium chloride aerosols. They found that the PNC measured by TSI NanoScan SMPS model 3910
is within 13% of the reference MPSS for monodisperse aerosols. However, to use these portable instruments in
ambient conditions, to the best of our knowledge, no such intercomparison study is available.
The goal of this study was to determine the uncertainties of PNCs and PNSDs measured by the TSI NanoScan SMPS
model 3910 and the GRIMM mini WRAS spectrometer 1371 portable particle size spectrometers in comparison to
reference MPSS and CPC of the WCCAP. We tested the portable instruments' performance and uncertainties using
certified monodisperse PSL particles, ambient urban aerosol, and a polydisperse sodium chloride aerosol.

## 2. Methodology:

### 2.1 Instrumentation

Two types of portable particle size spectrometers are compared against reference instrumentation (see Table 1). A
TROPOS-designed MPSS (referred to as WCCAP MPSS) served as a reference instrument for PNSD measurements.
It is regularly calibrated for sizing (PSL certified standard at 203 nm) and total particle number concentration, using
a calibrated reference CPC. The total CPC of the MPSS is regularly calibrated at the WCCAP against a calibrated
faraday cup aerosol electrometer (FCAE), which is annually calibrated at the PTB (Physikalisch-Technische
Bundesanstalt), the German National Metrology Institute (NMI). The MPSS and its calibration procedures are
described extensively in Wiedensohler et al. (2018).

### 2.1.1 TSI NanoScan SMPS model 3910

The TSI NanoScan SMPS model 3910 (TSI Inc., Shoreview, MN, USA) is a portable MPSS (Tritscher et al., 2013)
of compact dimensions (45 x 23 x 39 cm). It is specifically designed to measure PNSD within the range of 10-420 nm
(13 size channels while in scanning mode) with a sampling time of 60 s. A non-radioactive unipolar diffusion charger



(corona jet type; Medved et al. 2000), a radial differential mobility analyzer (rDMA; Zhang et al. 1995; Fissan et al.
1998), and an isopropanol-based CPC are the main components of this instrument. The working principle is as follows.
The aerosol flow (inlet: 0.75 L min$^{-1}$) enters the instrument and is then pre-conditioned to remove larger particles
using a cyclone with a cut-off diameter of 550 nm. Afterwards, all aerosol particles are positively charged in a corona-
jet-type unipolar diffusion charger using the opposed flow technique to ensure the stability of the ionizer needle. The
0.25 L min$^{-1}$ of the charged aerosol sample flow passes through a radial DMA, whose bottom plate is at a high negative
voltage and the top is at ground. During 45 s of the 'scanning mode' measurement, the radial DMA's voltage is ramped
up to scan the particle size range from 10 to 420 nm (equivalent mobility diameter in case of singly charged particles).
The particles are counted in an isopropanol-based CPC. This built-in CPC is similar to the handheld CPC model 3007
(TSI Inc.) (Hameri et al., 2002). Applying an inversion matrix including a unipolar charge distribution, the PNSD is
calculated with a size resolution of 13 size bins (midpoint diameters are: 11.5, 15.4, 20.5, 27.4, 36.5, 48.7, 64.9, 86.6,
115.5, 154.0, 205.4, 273.8 and 365.2 nm). From the inverted PNSD the instrument determines and reports the total
PNC and geometric mean diameter as well.

### 2.1.2 GRIMM Mini WRAS spectrometer 1371

The GRIMM Mini WRAS spectrometer 1371 (Grimm Aerosol Technik) is also a compact device for aerosol
measurements (23 x 25 x 22 cm) that combines two measurement techniques: an optical aerosol spectrometer to
determine the particle size distribution in 31 equidistant channels from 250nm to 35µm and an electric sensor called
"nano sizer" to size ultrafine particles by their electrical mobility diameter in the size range from 10 to 200 nm with a
resolution of 10 size bins (midpoint diameters are: 10, 14, 19, 27, 37, 52, 72, 100, 139, 193 nm). Details on the GRIMM
optical aerosol spectrometer is reported e.g. by Burkart et al., (2010). The nano sizer consists of a unipolar diffusion
charger, a deposition electrode, and an FCAE. Here, the aerosol inlet flow rate of 1.2 L min$^{-1}$ is led to a unipolar
diffusion charger. This charger generates a high ion number concentration using high positive voltage between a
central corona wire and a surrounding circular screen grid. The ions are then accelerated by the electric field in the
direction of the screen, pass it, and are directed further towards the outward-lying grounded housing (virtual earth).
The sample aerosol is passed through the ion cloud between the screen grid and the grounded housing, and the aerosol
particles are unipolarly charged. Subsequently, the particles enter the deposition electrode, where a negative voltage
is continuously ramped in 10 steps from high voltage to low voltage within 60 seconds, thereby changing the threshold
electrical mobility of particles that are allowed to enter the FCAE for detection. The PNSD is calculated by using an
inversion algorithm that includes Kernel functions for the size-dependent penetration efficiency of charged,
monodisperse particles through the deposition electrode.

Table 1: Specifications of instruments used during the inter-comparison workshop. Instruments No. 1 and 2 are the
portable aerosol spectrometers under study, while No. 3 and 4 are WCCAP's reference instrumentation.

| | Instrument | Manufacturer | Studied Metric | Size range (nm) | Size resolution (Total number of bins) | Time resolution (s) | Aerosol/Inlet Flow (L min$^{-1}$) | Sheath flow (L min$^{-1}$) | Other Specifications |
|---|---|---|---|---|---|---|---|---|---|
| 1. | TSI NanoScan SMPS model 3910 | TSI Inc. | PNSD+PNC | 10-420 | 13 | 60 | 0.75 | - | Non-radioactive, unipolar diffusion charger (corona jet type) |
| 2. | GRIMM Mini WRAS model 1371 | GRIMM Aerosol Technik | PNSD+PNC | 10-193 | 10 | 60 | 1.2 | - | Non-radioactive, unipolar diffusion charger, Faraday |





| | | | | | | | | Cup Aerosol Electromete r (FCAE) |
|---|---|---|---|---|---|---|---|---|
| 3. | WCCAP MPSS | WCCAP | PNSD+PNC | 10-800 | 71 | 300 | 1 | 5 | Bipolar diffusion charger, $^{85}$Kr, 370 MBq radioactive source, TSI CPC 3772 |
| 4. | Reference CPC model TSI 3772 | TSI Inc. | PNC | > 10 nm | - | 1 | 1 | - | - |


**2.2 Laboratory setup and Experimental approach:**

The intercomparison experiments of the portable instruments were divided into two periods: NanoScan SMPS model 3910 from Jan. 27-29, 2020, and the GRIMM Mini WRAS spectrometer 1371 from Jan. 29-31, 2020. Data were recorded for 1 min average for the portable instruments, while for WCCAP MPSS, the 5 min averaged data was generated. Most of the participating TSI NanoScan SMPS model 3910 were operated with NanoScan Manager version 1.0, while the FMI instruments had a homemade data acquisition software and the firmware 1.2 and 1.3 for their two instruments, respectively (Table A1).


Table 2: Specifications of instruments used during the inter-comparison workshop.

| Instruments | Participating Institutes |
|---|---|
| TSI NanoScan | 1. TSI GmbH, Germany<br>2. Technische Universität Braunschweig, TUBS<br>3. Danish Technological Institute, DTI<br>4. Institute for Combustion and Power Plant Technology, IFK<br>5. Institute of Environmental Assessment and Water Research, IDAEA-CSIC<br>6. Wessling GmbH<br>7. Norwegian University of Science and Technology, NTNU<br>8. Finnish Meteorological Institute, FMI<br>9. Politecnico di Torino, PdT<br>10. Federal Environment Agency (UmweltBundesamt), UBA Langen |
| GRIMM Mini WRAS | 1. Università Cattolica del Sacro Cuore, UNICATT<br>2. GRIMM Aerosol Technik<br>3. Institute of Ceramic Technology, ITC |
| WCCAP MPSS | 1. Leibniz Institute for Tropospheric Research, TROPOS |


Table 3: Experimental procedure followed during the inter-comparison workshop.

| Date | Experimental activities from January 27-31, 2020 |
|---|---|
| January 27, 2020 | Initial intercomparison without service and maintenance (TSI NanoScan SMPS)<br>➢ Setting up the TSI NanoScan SMPSs beside the WCCAP MPSS.<br>➢ Zero and leak check<br>➢ Flow checks using Gillian Gilibrator. |





|  | ➢ Overnight run (initial intercomparison) of all TSI NanoScan SMPS instruments and the WCCAP MPSS from 06.00 pm- 06.00 am, using the ambient aerosol. |
| --- | --- |
| January 28, 2020 | Final intercomparison after service and maintenance (TSI NanoScan SMPS)<br>➢ A manufacturer's maintenance service.<br>   ● Cleaning of the inlet impactor<br>   ● Checking or exchanging of the wick and filters<br>   ● Cleaning of the unipolar charger and cyclone<br>➢ Size calibration with certified 125 nm PSL particles<br>➢ Overnight run (final intercomparison) of all TSI NanoScan SMPS instruments and the WCCAP MPSS from 06.00 pm to 06.30 am, using the ambient aerosol |
| January 29, 2020 | ➢ Size calibration with certified 125 nm PSL particles (TSI NanoScan SMPS)<br>➢ Zero and leak check<br>➢ Flow checks using Gillian Gilibrator<br>➢ Calibration with polydisperse NaCl particles. |
| January 29, 2020 | Initial intercomparison without service and maintenance (GRIMM Mini WRAS)<br>➢ Setting up the GRIMM Mini WRAS beside the WCCAP MPSS.<br>➢ Zero and leak check<br>➢ Flow checks using Gillian Gilibrator.<br>➢ Overnight run (initial intercomparison) of all GRIMM Mini WRAS instruments and the WCCAP MPSS from 06.00 pm- 06.00 am, using the ambient aerosol (UNICATT instrument run on software version 10.0, ITC used version 7.2 instrument model while both GRIMM instruments were operated with on 8.2 version; Table A4) |
| January 30, 2020 | Final intercomparison after service and maintenance (GRIMM Mini WRAS)<br>➢ Size calibration with certified 125 nm PSL particles<br>➢ Calibration with polydisperse NaCl particles<br>➢ All GRIMM Mini WRAS are changed to software version 10.0<br>Overnight run (final intercomparison) of all GRIMM Mini WRAS instruments and the WCCAP MPSS from 06.00 pm-06.30 am, using the ambient aerosol |



### 3. Results and Discussion

### 3.1 Initial inter-comparison of the NanoScan SMPS model 3910 instruments using ambient aerosol

The first intercomparison experiment was done with all instruments without any service to determine the actual performance at the arrival. The intercomparison was done from 06.00 pm on Jan. 27 to 06.00 am on Jan. 28, 2020, using the WCCAP MPSS as a reference. During ambient aerosol sampling, the NTNU instrument failed just after 10 minutes of operation and sampled room-air. Thus, NTNU data is not considered in figures 1 and 2. Based on the contour plot of the PNSD, the most stable time periods were selected for discussion and interpretation. Figure 1a represents the intercomparison for the ambient run (Jan. 27, 2020) from 07.00 pm to 11.00 pm. During this period, mainly a bimodal PNSD in ultrafine aerosol mode was observed. However, NanoScan instruments failed to identify the peak around 25 nm. Compared to the WCCAP MPSS, the mode peak in ultrafine aerosol mode for NanoScan SMPS was deviated by 10% in size. Furthermore, the PNC of the first ultrafine aerosol mode was underestimated by 60%, and the PNC of dominant ultrafine aerosol mode was overestimated by 120%. The latter is probably a misclassification caused by the unipolar charging. The PNC of the NanoScan SMPS instruments lies mostly within the ±20% range of the PNC measured by the reference CPC as shown in Fig. 1b. Additional uncertainties for the PNSD and PNC may also derive from the limited number of particle size bins, as described in (Buonanno et al., 2009).



Furthermore, the NanoScan SMPS showed different PNSD for the size range greater than 200 nm and an
underestimation of the total PNC by 80%.

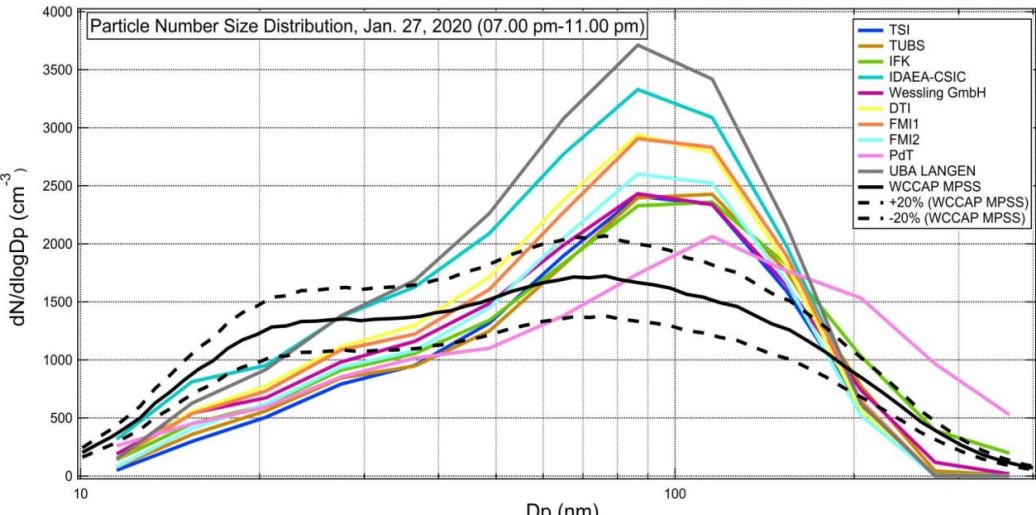



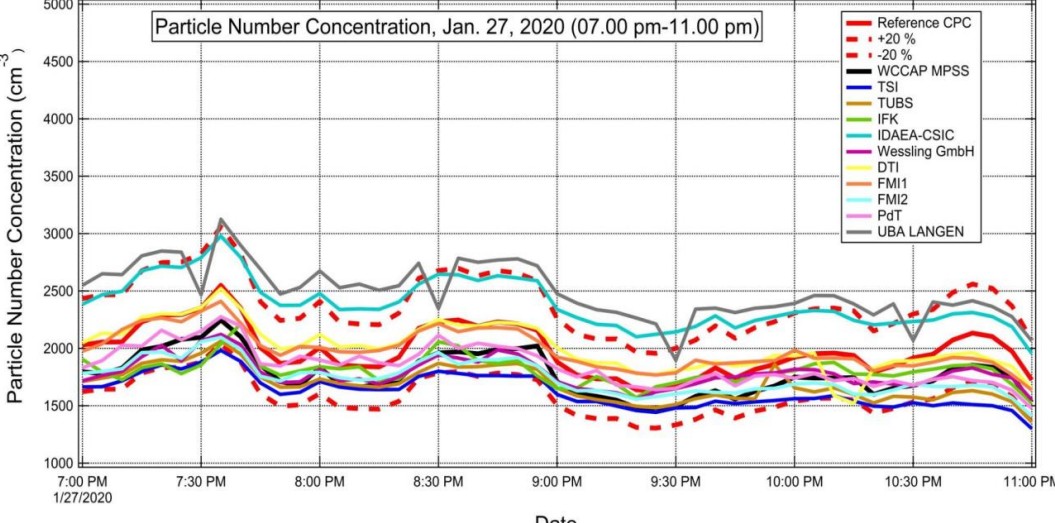


Figure 1: PNSD ambient intercomparison of the NanoScan SMPS model 3910 instruments on Jan. 27, 2020 from
07.00 pm to 11.00 pm. The dashed black lines show ±20% range in sizing (a) The dark black solid line shows the
PNSD of the WCCAP MPSS. (b) Time series of the PNCs. The PNC of the reference CPC is represented by the solid
red line, while the red dotted lines show the ± 20% range. The solid black line represents the integrated PNC of the
WCCAP MPSS.






**3.2 Final inter-comparison of the NanoScan SMPS model 3910 after service and maintenance**
**3.2.1 Size calibration NanoScan SMPS model 3910 with PSL particles**
The size calibrations were performed using certified PSL (polystyrene latex) particles of 125 nm. This PSL particle
size was used for two reasons (1) in a dilute solution, the number concentration of PSL particles is sufficiently high
(1 drop i.e. 1% by volume in 150 ml pure water) (2) for particle size larger than 100 nm, residual material layer from
aqueous solution on PSL particles is not significant (Wiedensohler et al., 2018). Figure 2 shows that a TSI NanoScan
SMPS model 3910 cannot resolve the monodisperse peaks of single and doubly charged PSL particles due to the
limited size resolution.

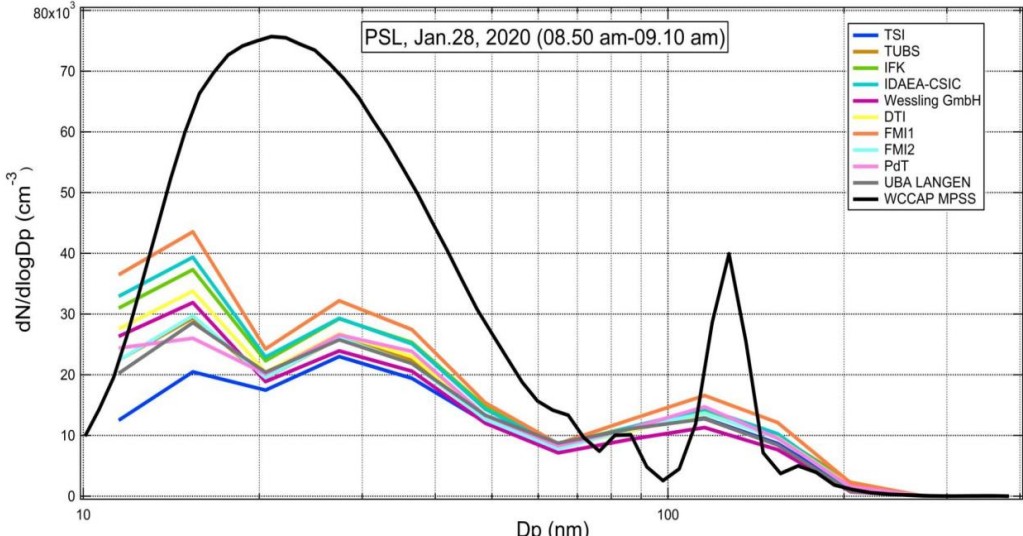


Figure 2 Size calibration of the NanoScan SMPS model 3910 with 125 nm certified PSL particles. The closest size
bin is at 115.5 nm for the NanoScan instrument as compared to PSL peak. The solid black line shows the PSL
calibration of the WCCAP MPSS.











**3.2.2    Inter-comparison of the NanoScan SMPS model 3910 using ambient aerosol**

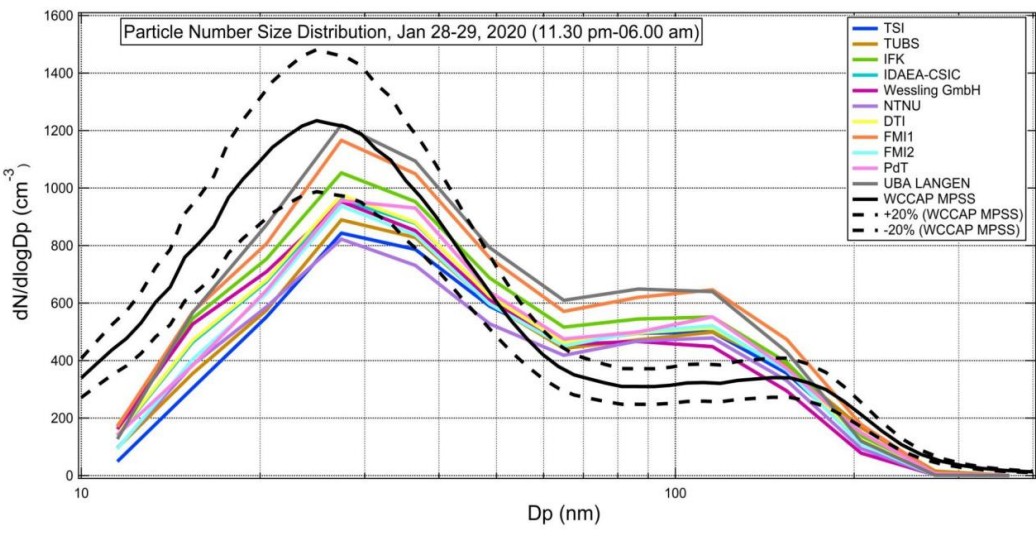


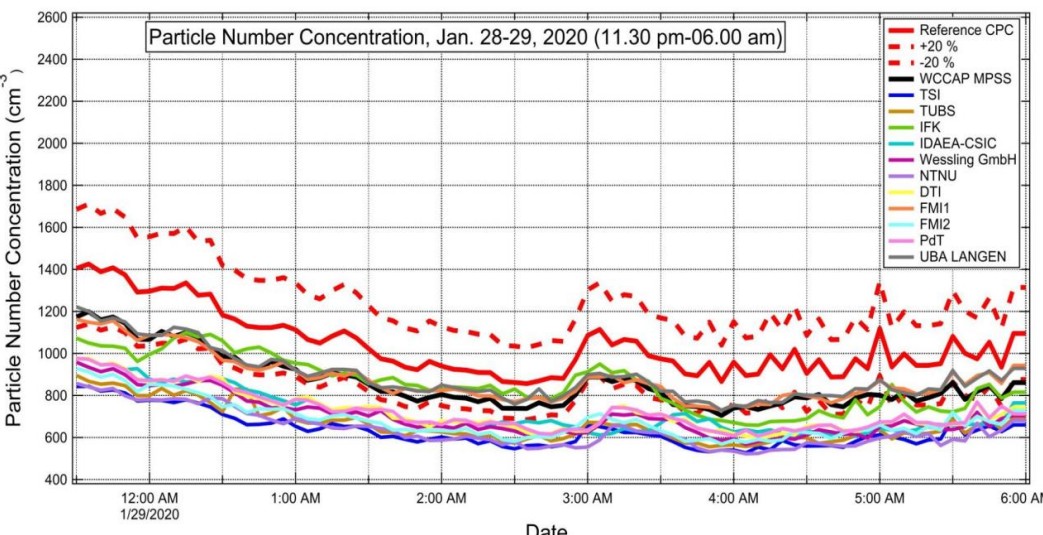


Figure 3: (a) PNSD ambient intercomparison of the TSI NanoScan SMPS instruments on Jan. 28-29, 2020 from 11.30
pm to 06.00 am. The black solid line shows the PNSD of the WCCAP MPSS.  The dashed black lines show ±20%
range in sizing (b) Time series of the PNC. The PNC of the reference CPC is represented by the solid red line, while
the red dotted lines show the ±20% range.  The solid black line represents the integrated PNC of the WCCAP MPSS.
Based on the results shown in Figure 3a, the performance of all NanoScan SMPS model 3910 was found to be
significantly improved after service and maintenance. Figure 3a represents the intercomparison for the ambient run
(Jan. 28-29, 2020) from 11.30 pm to 06.00 am. During this period, mainly a bimodal PNSD was observed. The TSI
NanoScan SMPS instruments underestimate the PNC in the ultrafine aerosol mode by up to 40% compared to the
WCCAP MPSS. The mode peak deviations in the ultrafine aerosol mode was approximately 10% compared to the
mode peak diameter of MPSS. The PNC measured by NanoScan SMPS were overestimated up to 80% when compared





with WCCAP MPSS in the fine aerosol mode. The latter result seems to be a systematic effect of the unipolar charging
and the reduced sensitivity of the electrical particle mobility with an increasing particle size above 200 nm. There is a
slight shift in distribution observed for NanoScan instruments. This could be due to the algorithm limitation as with
bimodal distribution the inversion matrix reaches its limit. In Figure 3b, the integrated PNC of the WCCAP MPSS
was within the ±20% range, while most of the NanoScan SMPS model 3910 were within the 20-40% range as
compared to the reference CPC. Here, a reasonably good agreement was found between unit-to-unit (i.e. within the
±20% range).
**3.2.3 Calibration of TSI NanoScan SMPS model 3910 using a polydisperse NaCl aerosol**

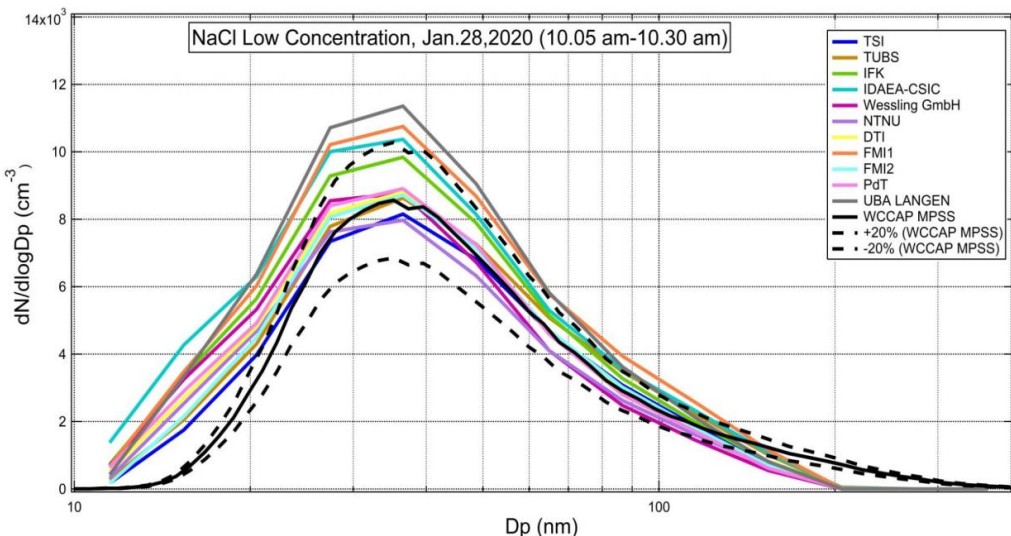


Figure 4: Performance of NanoScan SMPS model 3910 using a nebulizer-generated NaCl aerosol with PNC of
approximately 10,000 cm$^{-3}$. The solid black line shows the WCCAP MPSS.
The last step of the calibration of the NanoScan SMPS model 3910 was to use a polydisperse unipolarly pre-charged
nebulizer-generated laboratory aerosol in the size range below 100 nm. In Figure 4, the peaks of the PNSDs at
approximately 35 nm measured by NanoScan SMPS instruments agree well with the WCCAP MPSS. The sizing
accuracy of most of the NanoScan SMPS instruments is within ±20% uncertainty range except for two instruments.
The two units overestimated the PNC by 25% and 30% respectively from WCCAP MPSS. The inversion matrix is
calibrated by monomodal particles so the algorithm behaves reasonably well.









**3.3 Initial inter-comparison of the GRIMM Mini WRAS spectrometer 1371 without service and maintenance**

Figure 5: (a) PNSD ambient intercomparison of the GRIMM Mini WRAS spectrometer 1371 on Jan. 29-30, 2020 from 11.00 pm to 06.00 am. The dashed black lines show ±20% range in sizing (b) Time series of the PNC. The PNC of the reference CPC is represented by the solid red line, while the red dotted lines show the ±20% range. The solid black line represents the integrated PNC of the WCCAP MPSS.

Figure 5a represents the ambient intercomparison on Jan. 29-30, 2020 from 11.00 pm to 06.00 am. Here, a bimodal PNSD was observed with the WCCAP MPSS. The dominating ultrafine aerosol mode peak was observed for the UNICATT instrument operating with the software version 10.0. The ultrafine aerosol mode PNC for the UNICATT instrument was within ±20% range compared to the WCCAP MPSS. For the other GRIMM Mini WRAS spectrometers (i.e. ITC and GRIMM) operating with software version 7.2 and 8.2 respectively, the ultrafine aerosol mode peak deviation from the WCCAP MPSS was 56% while the ultrafine aerosol mode PNC was underestimated by 40%. The fine aerosol mode peak around 180 nm could not be resolved by all instruments irrespective of the



software used. The difference between the software version lies in different inversion matrices. In Figure 5b, the PNC
were compared and only the UNICATT instrument operating with software version 10.0 remains within ±20% range
compared to the PNC of the reference CPC. The PNC was underestimated by 60% by other instruments when
compared to the PNC of the reference CPC.
**3.4 Final inter-comparison of the GRIMM Mini WRAS spectrometer 1371 after service and maintenance**
**3.4.1 Size calibration of GRIMM Mini WRAS spectrometer 1371 with PSL particles**

PSL, Jan. 30, 2020 (12.15 pm-12.30 pm)

Legend:
UNICATT
ITC
GRIMM1
GRIMM2
WCCAP MPSS

Y-axis: dN/dlogDp (cm⁻³), from 0 to 50×10³
X-axis: Dp (nm), from 10 to 100


Figure 6: Size calibration of the GRIMM Mini WRAS spectrometer 1371 with 125 nm certified PSL particles. The
closest size bin is at 139 nm for the Mini WRAS instrument as compared to PSL peak. The black line shows the PSL
calibration of the WCCAP MPSS.
Figure 6 shows that a GRIMM Mini WRAS spectrometer 1371 cannot resolve the monodisperse peaks of single and
doubly charged PSL particles due to the limited size resolution. The UNICATT instrument showed a different behavior
when challenged with PSL particles than other GRIMM Mini WRAS spectrometers 1371. This could be due to
software version 10 used by UNICATT while the rest other used old software versions.











### 3.4.2 Inter-comparison of GRIMM Mini WRAS spectrometers using ambient aerosol

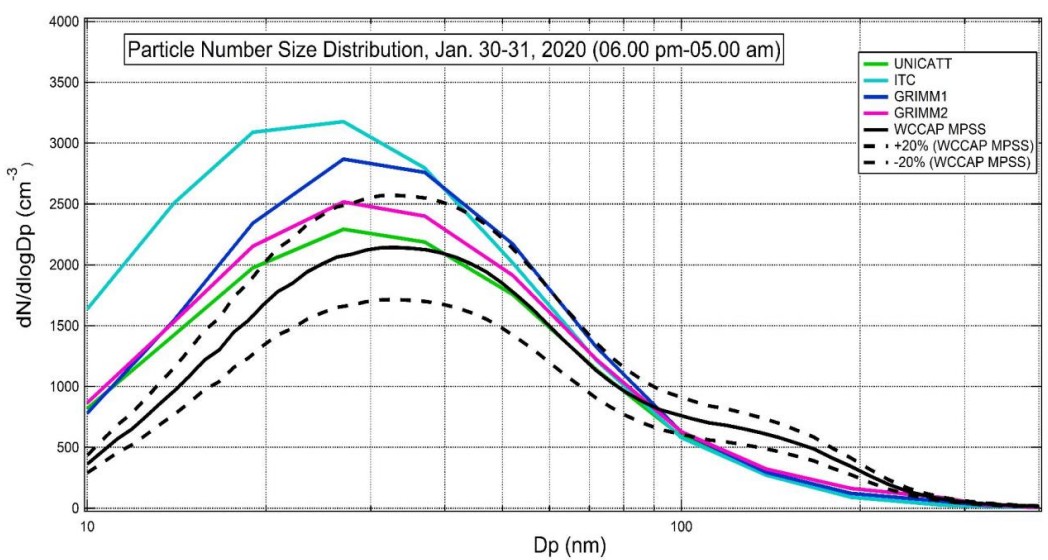

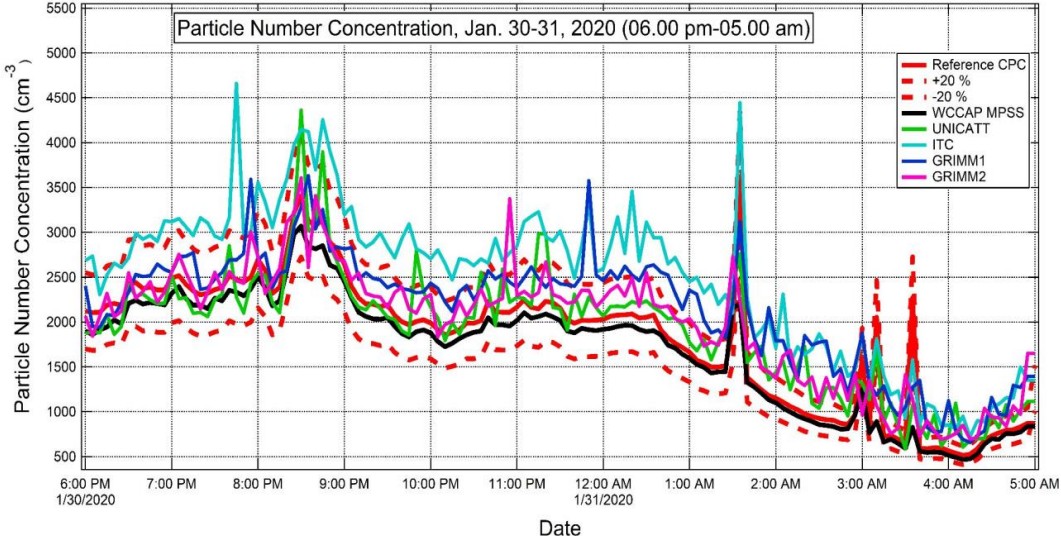

Figure 7: (a) PNSD ambient intercomparison of the GRIMM Mini WRAS spectrometer 1371 on Jan. 30 to 31, 2020 from 06.00 pm to 05.00 am. The solid black line shows the PNSD of WCCAP MPSS. The dashed black lines show ±20% range in sizing (b) Time series of the PNC. The PNC of the reference CPC is represented by the solid red line, while the red dotted lines show the ±20% range. The solid black line represents the integrated PNC of the WCCAP MPSS.

All the four GRIMM Mini WRAS spectrometers 1371 were operated with software version 10.0. It needs to be pointed out that operating the Mini WRAS with software version 10.0 requires instrument-specific calibration factors that were only available for the GRIMM Mini WRAS spectrometer 1371 of "UNICATT" during the calibration workshop. The other GRIMM Mini WRAS spectrometers 1371 were operated with "default" values for the calibration factors. Therefore, larger deviations from the results of the reference instrument need to be expected. Figure 7a, representing





the ambient intercomparison, showed a dominant ultrafine aerosol mode peak around 35 nm. The GRIMM Mini
WRAS spectrometer 1371 deviated by 16% in the mode peak diameter in ultrafine aerosol mode while the PNC of
the ultrafine aerosol mode of all instruments was overestimated between 10-50%. All GRIMM Mini WRAS (operating
with software version 10.0) overestimated the PNC between 10 and 50% when there was a dominant ultrafine aerosol
mode. The fine aerosol mode peak around 130 nm could not be not detected and PNC of fine aerosol mode was
systematically underestimated above 100 nm by 60%. Figure 7b, representing the integrated PNC when compared
with the reference CPC. Except for instruments from ITC, all other GRIMM Mini WRAS spectrometers were within
±20% uncertainty range. The GRIMM Mini WRAS spectrometer 1371 performance was found to be improved after
cleaning, & servicing as well when operated with the software version 10.0.
**3.4.3 Calibration of the GRIMM Mini WRAS spectrometer 1371 using a polydisperse NaCl aerosol**

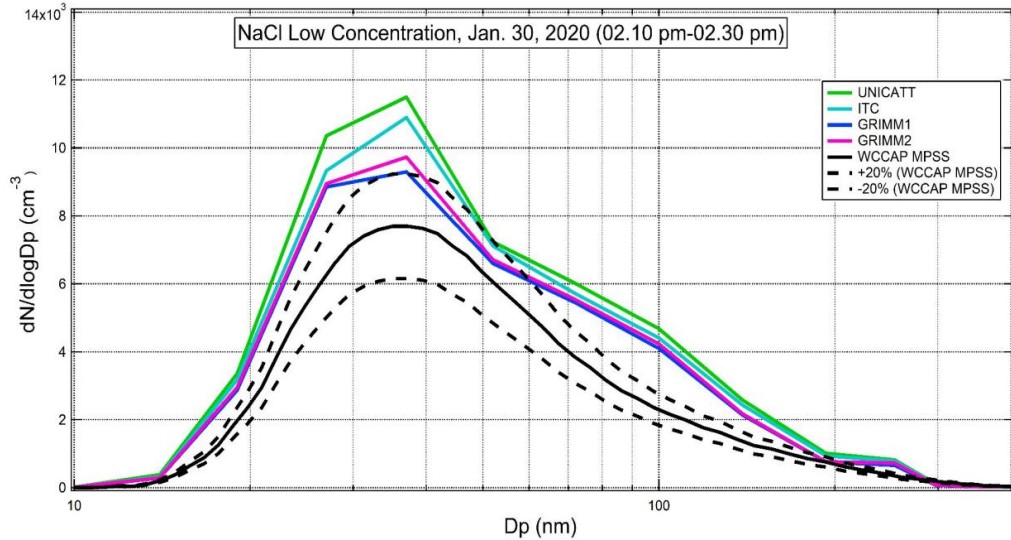


Figure 8: Performance of the GRIMM Mini WRAS spectrometers 1371 using a nebulizer-generated NaCl aerosol
with PNC of approximately 10,000 cm$^{-3}$. The black dotted line shows the WCCAP MPSS.
In Figure 8, the peak of PNSDs at approximately 35 nm measured by GRIMM Mini WRAS spectrometers 1371 agree
well with the WCCAP MPSS in terms of mode peak. The agreement looks good when the mode peak is compared
while the size distribution measured by most of the GRIMM Mini WRAS spectrometers 1371 misses the ±20%
uncertainty range compared to WCCAP MPSS. The GRIMM Mini WRAS instruments overperformed by 20-40%
when compared with the WCCAP MPSS. The algorithm behaves reasonably well as the inversion matrix is calibrated
by monomodal particles.
In addition, the inversion matrix of software version 10.0 created an artificial peak around 100 nm.
**3.5. Performance of the WCCAP MPSS and reference CPC**
The following plots show the correlation of the integrated PNC of the WCCAP MPSS versus the PNC measured by
the reference CPC. Figures 9 a, b, c, and d show an underestimation of the MPSS derived PNC between 10-15% for
different time period.



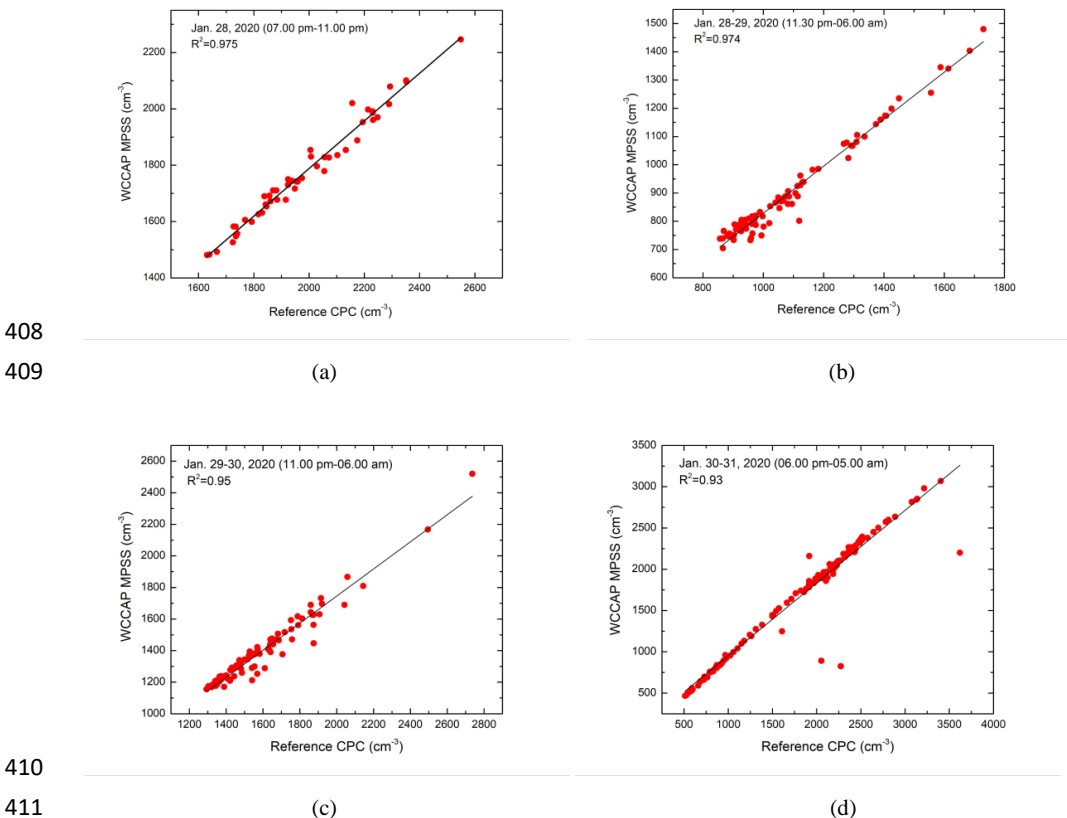


409                                 (a)                                                            (b)


411                                 (c)                                                            (d)

Figure 9: Correlation of the PNC of the WCCAP MPSS versus the reference CPC for the ambient intercomparison
periods: (a) Jan. 27, 2020 (b) Jan. 28-29, 2020 (c) Jan. 29-30, 2020 (d) Jan. 30-31, 2020.

## 4. Summary and recommendations

The performance of portable MPSS, the NanoScan SMPS model 3910, and the GRIMM Mini WRAS spectrometer
1371 were evaluated in intercomparison workshops against a reference MPSS and CPC of the WCCAP. Inter-
comparison and calibrations with ambient and laboratory-generated aerosols respectively were performed at the
WCCAP, Leipzig, Germany from Jan. 27-31, 2020.
The following general recommendations are important for the TSI NanoScan SMPS model 3910 and GRIMM Mini
WRAS 1371 spectrometers based on workshop results:
-    It is important to clean and service the instruments on a yearly basis to improve their performance. It is
424         advised that users should carry out such activities at their own institute/facilities. This includes the cleaning
425         of various parts such as inlet impactor, wick, filter check, cleaning of cyclone and charger, etc.
-    It is recommended to run initial zero and leak checks in order to find any internal leak before the instrument
427         operation.
After service, cleaning and performing zero and leak checks, following performances have been identified:



TSI NanoScan SMPS:

- The performance of NanoScan SMPS instruments improved for the ultrafine aerosol mode while the PNC in the fine aerosol mode still overestimated by up to 80%. This is due to reduced sensitivity of electrical particle mobility with increasing particle size above 200 nm.

- The performance of some of the NanoScan SMPS found to be in good agreement (i.e. within 20%) compared to the reference CPC, considering integral PNC.

- The mode peak deviations (difference in peak diameter of NanoScan mode peak from WCCAP MPSS mode peak diameter) in the ultrafine aerosol mode was within limit i.e. approx. 10%. However, the peak height measured by NanoScan instruments is lower as compared to MPSS.
- A reasonably good unit-to-unit agreement within ±20% was found for NanoScan SMPS instruments.

GRIMM mini WRAS spectrometer:

- The performance of Mini WRAS spectrometer run with software version 10.0 found to be improved significantly with less uncertainties than the previous software versions 7.2 and 8 respectively, when compared to the WCCAP MPSS.
- The mode peak deviations (difference in peak diameter of Mini WRAS mode peak from WCCAP MPSS mode peak diameter) for ultrafine aerosol mode was 15%. However, the peak height measured by Mini WRAS instruments is higher as compared to MPSS.
- With dominant ultrafine aerosol mode, most of the GRIMM Mini WRAS instruments (operating with software version 10.0) agree well with PNC (i.e. 10-50%). Conversely, PNC of the fine aerosol mode was systematically underestimated by 60% above 100 nm due to limitation of the inversion matrix.

- Except for one instrument, the integral PNC of the GRIMM Mini WRAS spectrometers were within an uncertainty range of ±20% compared to the reference CPC.

Additional results:

- Calibrations were done with certified PSL particles of 125 nm and polydisperse laboratory-generated NaCl particles. Both the TSI NanoScan SMPS and the GRIMM Mini WRAS spectrometer 1371 were not able to resolve the monodisperse PSL particles due to the limited size resolution.
- Both, the NanoScan SMPS model 3910 and GRIMM Mini WRAS spectrometers 1371 were able to determine the peak diameter of a polydisperse unipolarly pre-charged nebulizer-generated NaCl aerosol in the size range below 100 nm.

This intercomparison study provided the advantages and limitations of both the portable instruments i.e. NanoScan SMPS and Mini WRAS. Based on the workshop result, these portable instruments are easy to use and are suited for mobile ultrafine particle measurements, especially to detect relative differences in the PNSD such as source apportionment studies of ultrafine aerosol particles at work places or outdoors near sources. These portable instruments can also be used for nanotechnology workplaces with appropriate care.

We recommended to users how best performance can be achieved using these portable instruments at workplaces or outdoor near sources based on inter-comparison workshop results. However, further field studies might be required to determine exactly how to apply these portable instruments for a good performance during mobile measurements when installed for example on backpacks or drones.

**Data availability.** The data can be made available upon request.

**Author contributions.** KW, WB and AW planned and designed the study. All co-authors participated in the experiments. AA processed the data and prepared the manuscript with inputs from WB, TT, GS, KW and AW. All of the co-authors proofread and commented on the manuscript.

**Competing Interests.** The authors declare no conflict of interest.





**Acknowledgements.** This investigation was supported by the Umweltbundesamt in the frame of the project
"Fortsetzung des World Calibration Centers for Aerosol Physics (WCCAP) im Rahmen des GAW-Programms (Global
Atmosphere Watch) der WMO Genf (2019-2022)" with the project number 113833.

**Appendix A: Tables consisting of technical details of portable instruments during intercomparison experiment**
Table A1. Technical details of the TSI NanoScan SMPS instruments Day 1 (Jan.27, 2020)

| Serial Number | Owner | DAQ Software and Version | Last Calibration | Last filter/ wick change | inlet flow measured (L min$^{-1}$) Day 1 | inlet flow displayed (L min$^{-1}$) Day 1 | other info |
|---|---|---|---|---|---|---|---|
| 3910181009 | TSI | Device internal, NanoScan Manager 1.0 | NA | NA | 753.7 AM, 749.1 PM | n/a AM, 764 PM | |
| 3910122701 | Technische Universität Braunschweig, TUBS | Device internal, NanoScan Manager 1.0 | July 31, 2012 | Jan.20, 2020 | 725 (AM)/ 709.2 (PM) | 714 | |
| 3910151401 | Danish Technological Institute, DTI | Device internal, NanoScan Manager 1.0 | Jan. 8, 2019 | Probably at calibration | 684.9 | 684 | |
| 3910174404 | IFK | Device internal, NanoScan Manager 1.0 | May 22, 2018 | NA | 726 | 698 | |
| 3910131603 | IDAEA-CSIC | Device internal, NanoScan Manager 1.0 | March 1, 2018 | NA | 810.1 | 823 | Laser current error (mA): 67.9 (11:19) |
| 3910161701 | Wessling GmbH | Device internal, NanoScan Manager 1.0 | Nov. 16, 2018 | Probably at calibration | 696 | 742 | |
| 3910164102 | Norwegian University of Science and Technology, NTNU | Device internal, NanoScan Manager 1.0 | Sept. 20, 2019 | Probably at calibration | 717 | 749 | |



| 3910141702 | Finnish Meteorological Institute, FMI | Homemade, firmware 1.2 | April 30, 2019 | Jan. 27, 2020 | 785 | 782 | |
|---|---|---|---|---|---|---|---|
| 3910154701 | Finnish Meteorological Institute, FMI | Homemade, firmware 1.3 | Feb. 18, 2019 | Jan. 27, 2020 | 692 | 723 | |
| 3910151403 | Politecnico di Torino (PdT) | Device internal, NanoScan Manager 1.0 | Nov. 14, 2017 | NA | 745.5 | 739 | |
| 3910182301 | Umweltbundesamt Langen | NanoScan Manager 1.0 | Jan. 1, 2018 | June 22, 2018 | 760.5 at 12:30; 740.7 at 15:00 | 794 | |



Table A2. Technical details of NanoScan SMPS instruments from 10 different institutes (Jan. 28, 2020) after servicing and maintenance.

| Device | Serial Number and Owner | inlet flow day2 (ccm) - measured | inlet flow day2 (ccm) - instrument | $Q_{inlet, measured/displayed}$ | Service done during workshop | inlet flow day2 (ccm) - measured after servicing | inlet flow day2 (ccm) - instrument after servicing | $Q_{inlet, measured/displayed}$ | other info |
|---|---|---|---|---|---|---|---|---|---|
| NanoScan SMPS | 3910181009 (TSI) | 745 | 760 | 0.98 | Only the inlet impactor was cleaned. Checked wick. | 746.3 | 763 | 0.98 | |
| NanoScan SMPS | 3910122701 (TUBS) | 703.2 | 750 | 0.94 | inlet cleaned, wick changed | 705.4 | 710 | 0.99 | |
| NanoScan SMPS | 3910151401 (DTI) | 680.1 | 674 | 1.01 | Inlet cleaned, wick filter changed, charger cleaned, cyclone cleaned | 681.5 | 702 | 0.97 | |
| NanoScan SMPS | 3910174404 (IFK) | 722 | 704 | 1.03 | Inlet cleaned, wick filter changed, charger cleaned, cyclone cleaned | 717.1 | 720 | 1.00 | |
| NanoScan SMPS | 3910131603 (IDAEA-CSIC) | 786.2 | 820 | 0.96 | inlet cleaned, wick checked | 793.9 | 828 | 0.96 | Laser current error (mA): 67.9 (10:45) |





| Device | Serial Number | | | | | | | | |
|---|---|---|---|---|---|---|---|---|---|
| NanoScan SMPS | 3910161701 (Wessling GmbH) | 695.3 | 743 | 0.94 | inlet cleaned, wick new, IPA new, 2 internal small filters new, two tubes new | 680.7 | 722 | 0.94 | |
| NanoScan SMPS | 3910164102 (NTNU) | 717 | 714 | 1.00 | inlet cleaned, wick was checked | 716 | 720 | 0.99 | |
| NanoScan SMPS | 3910141702 (FMI 1) | 774 | 777 | 1.00 | cyclone and charger cleaned, wick changed, changed filters, cut tubing ends to make them tighter, checked cpc performance by using an inline filter: zero check still fails, needs service | 783 | 782 | 1.00 | |
| NanoScan SMPS | 3910154701 (FMI 2) | 690 | 721 | 0.96 | cyclone and charger cleaned, wick and filters changed | 709 | 724 | 0.98 | |
| NanoScan SMPS | 3910151403 Politecnico di Torino (PdT) | 743.7 | 736 | 1.01 | Inlet cleaned, wick changed, charger cleaned, reservoir cleaned, one filter changed | 755.9 | 749 | 1.01 | |
| NanoScan SMPS | 3910182301 (UBA LANGEN) | 739.1 | 785 | 0.94 | inlet cleaned, wick changed, new pump (left one from looking left), 2 internal small filters | 754.7 | 814 | 0.93 | |



Table A3. Technical details of NanoScan SMPS instruments from 10 different institutes (Jan. 29, 2020).

| Device | Serial Number | Owner | inlet flow day3 (ccm) - measured | inlet flow day3 (ccm) - instrument | $Q_{inlet,measured/displayed}$ |
|---|---|---|---|---|---|
| NanoScan SMPS | 3910181009 | TSI | 748.7 | - | - |
| NanoScan SMPS | 3910122701 | TUBS | 711.3 | 716 | 0.99 |
| NanoScan SMPS | 3910151401 | DTI | 678.7 | 696 | 0.98 |
| NanoScan SMPS | 3910174404 | IFK | 721.4 | 720 | 1.00 |



| NanoScan SMPS | 3910131603 | IDAEA-CSIC | - | - | - |
|---|---|---|---|---|---|
| NanoScan SMPS | 3910161701 | Wessling GmbH | 675 | 721 | 0.94 |
| NanoScan SMPS | 3910164102 | NTNU | 721 | 740 | 0.97 |
| NanoScan SMPS | 3910141702 | FMI 1 | 780 | 783 | 1.00 |
| NanoScan SMPS | 3910154701 | FMI 2 | 698 | 709 | 0.98 |
| NanoScan SMPS | 3910151403 | Politecnico di Torino (PdT) | 751.6 | 748 | 1.00 |
| NanoScan SMPS | 3910182301 | UBA LANGEN | 753 | 814 | 0.93 |



Table A4. Technical details of Mini WRAS spectrometer instruments from 3 different institutes on (Jan. 29, 2020).
UNICATT instruments operated with software version 10.0 while ITC at version 7.2 and GRIMM instruments at
version 8.2.

| Device | Serial Number and Owner | DAQ Software Version | Last Calibration | Last filter/wick change | Rinsing Air flow (L min$^{-1}$) | Inlet flow day 3 ccm measured (L min$^{-1}$) | Charger status (nA) | High voltage of the corona charger (V) | other info |
|---|---|---|---|---|---|---|---|---|---|
| MiniWRAS | 71-16-06 UNICATT | ver. 10.0 | May 1, 2019 | May 1, 2019 | 0.549 | 1.205 | 2.5 | 3250 | Silica gel changed on December 2019 |
| MiniWRAS | 71-16-09 ITC | ver. 7.2 | Sept. 1, 2016 | never | 0.572 | 1.189 | 2.504 | 3780 (Limit) | Silica gel changed Jan-20 |
| MiniWRAS | 71-19-09 Grimm Aerosol Technik | ver. 8.2 Rev I | Jan. 28, 2020 | Jan. 15, 2020 | 0.561 | 1.193 | 2.503 | 2999 | New Unit |
| MiniWRAS | 71-18-11 Grimm Aerosol Technik | ver. 8.2 Rev I | Jan. 28, 2020 | Jan. 15,2020 | 0.585 | 1.204 | 2.501 | 3250 | Demo Unit |







Table A5. Technical details of Mini WRAS spectrometer instruments from 3 different institutes (Jan. 30, 2020). All four instruments worked on software version 10.0.

| Device | Serial Number | Owner | DAQ Software and Version | Last Calibration | Last filter/wick change | Rinsing Air flow (L min⁻¹) | Inlet flow day 4 ccm measured (L min⁻¹) |
|---|---|---|---|---|---|---|---|
| MiniWRAS | 71-16-06 | UNICATT | ver. 10.0 | May 1, 2019 | May 1, 2019 | 0.54 | 1.179 |
| MiniWRAS | 71-16-09 | ITC | ver. 10.0 | Sept. 1, 2016 | never | 0.566 | 1.179 |
| MiniWRAS | 71-19-09 | Grimm Aerosol Technik | ver. 10.0 | Jan. 28, 2020 | Jan. 15, 2020 | 0.561 | 1.189 |
| MiniWRAS | 71-18-11 | Grimm Aerosol Technik | ver. 10.0 | Jan. 28, 2020 | Jan. 15, 2020 | 0.61 | 1.194 |

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
