# Peer review of "Performance analysis of the NanoScan SMPS and the Mini WRAS Ultrafine Aerosol Particle Size Spectrometers"

_Atmospheric Measurement Techniques, 2022_

## Referee Comment (RC1)

Review to "Performance analysis of the NanoScan SMPS and the Mini WRAS Ultrafine Aerosol Particle Size Spectrometers"

The authors present performance evaluations of two portable instruments against reference instrumentation for the measurement of particle number size distributions (PNSD) and total particle number concentration (PNC) during the workshop conducted at the World Calibration Center for Aerosol Physics (WCCAP) in Leipzig, Germany, in January 2020. The performances and uncertainties of the NanoScan SMPS (TSI, model 3910), and the Mini WRAS (Grimm, model 1371) were investigated against the WCCAP Mobility Particle Size Spectrometers (MPSS) and Condensation Particle Counters (CPC) using ambient aerosols and lab-generated PSL and NaCl particles. The inter-comparisons were performed both before and after the service and maintenance, and recommendations of timely service, maintenance and calibration were proposed, which will be instructive to the users. The manuscript is easy to follow and fits the scope of Atmospheric Measurement Techniques. However, I feel that the authors could provide more detailed work to demonstrate how service and maintenance improve the performance of the instruments, which will serve as valuable guidance for both existing and potential users. The reviewer recommends accepting this manuscript after addressing the following comments.

**Major comments:**
1) The NanoScan SMPS and the Mini WRAS are portable and easy to use. But considering their inferiorities in both the time and size resolution, they may not be a great choice for mobile-platform measurement. Both the NanoScan SMPS and the Mini WRAS are not considered as fast (i.e., time resolution of 60 s). The new generation SMPS (e.g., TSI model 3938) can provide fast scan measurements (e.g., 15s and below, https://tsi.com/products/particle-sizers/scanning-mobility-particle-sizer-spectrometers/general-scanning-mobility-particle-sizer-(smps)-3938/).

2) If possible, I recommend the authors provide performance evaluations of the instruments based on their factory calibration, and analyze how the performance would change during long-term operations, such that the users could have a professional application note to follow.

3) The author claimed that the TSI NanoScan SMPS instruments were significantly improved after service and maintenance, based on the comparisons of ambient PNSD measurement before and after maintenance. However, it is noticeable that the ambient PNSD of the two measurement periods are quite different. If the comparison is demonstrated by relative errors (i.e., concentration ratios), I suspect the relative error may still be comparable.

4) By comparing Fig. 2 and Fig. 3, I am wondering why the TSI NanoScan SMPS failed to capture the size distribution of particles in the ultrafine mode from the PSL solution, but got reasonably good agreements when measuring ambient aerosols of the similar size range (i.e., 10 ~ 50 nm). Same for the GRIMM Mini WRAS (except for the one from UNICATT).

5) If possible, please clarify how the sizing-relevant parameters (e.g., flow and/or voltage) look like before and after maintenance, especially the NanoScan SMPS from FMI2 which behaves quite differently after maintenance. I think that may help guide the users on how often to service the instrument.

6) With respect to the WCCAP MPSS, the NanoScan SMPS underestimated the PNC in the ultrafine aerosol mode for the intercomparison of ambient measurement, but overestimated the PNC when testing the polydisperse NaCl particles. Do the authors have any explanations for that?

**Minor comments:**

1) Table 1 row 1: Please also specify the particle counting technique of the NanoScan SMPS.

---

## Referee Comment (RC2)

The paper contains a set of measurements of several units of two models of portable self-contained nanoaerosol classifier-quantifier. The devices are evaluated against a reference instrument before and after servicing. Measurements against ambient aerosols and laboratory aerosols are performed in the size range of 10-200nm. Additional information regarding flowrate calibration, laser performance and time since last service is also included but not commented on. The comparison among various instruments of the same model is uncommon in the nanoaerosol field and relevant for future publications, especially after long usage intervals. The reviewer recommends addressing the following comments prior to publication.

- **Annual calibration:** "The results indicate that the portable instruments must be serviced and calibrated annually" The claim for this conclusion is not demonstrated. The flowmeter error and counting inaccuracy is measured for units with known last services, but (time-since-last-service vs counting-error[corrected for final inaccuracy]) is not displayed, therefore the statement "annually" has no basis.
- **Time series ambient intercomparison:** The methodology and usefulness for this measurement is unclear. I understand that the evaluated parameter is the stability over time of the overall counts. Additionally, since concentrations change over time, linearity can be evaluated Figures 9 a)b)c)d). The measurement method is confusing though. Particle number concentration over time incurs in the size dependent counting error. The Nanoscan SMPS ambient test, before and after the cleaning shows, two clearly different particle distributions. A more relevant representation should evaluate the time dependent ambient aerosol per size channel or at a set of size channels, this way time-linearity can be appropriately evaluated. Otherwise, if the ambient aerosol were to change its distribution counting will be affected. Figures 1b and 3b as printed, suggests that you should not service your device as you will loose counting efficiency.
- **Nebulizer-generated NaCl aerosol**: It is not specified the generation device/method for the precharged NaCl aerosol nor if the charger on the particle classifiers was turned off.
- There is little to no scientific value in including results from inaccurate or defective devices with regard to the GRIMM Mini WRAS spectrometers with SW <10 being uncalibrated. Unless a dedicated section addresses the background of said updates commenting on the inversion algorithm I would suggest removing any measurement data regarding said units. Data could be left as is, as a correction for previous publications with these instruments, if a comment is added.
- Regarding the test against 125nm PSL particles. This is quite a relevant plot, it provides a high concentration high resolution aerosol, which challenges both devices and shows the limitations of the instruments. For any of the cases it is not commented on the nature of the particles with aerodynamic diameters <<125nm. For 125nm the charging efficiencies of radioactive and corona should be similar but there is clearly a difference on the small size residues? A comment will help the reader with this plot. Maybe the devices did reach saturation

---

## Author Comment (AC1)

RR: Reviewer's response

AR: Author's response

Reviewer 1 comments:

RR:

Review to "Performance analysis of the NanoScan SMPS and the Mini WRAS Ultrafine Aerosol Particle Size Spectrometers"

The authors present performance evaluations of two portable instruments against reference instrumentation for the measurement of particle number size distributions (PNSD) and total particle number concentration (PNC) during the workshop conducted at the World Calibration Center for Aerosol Physics (WCCAP) in Leipzig, Germany, in January 2020. The performances and uncertainties of the NanoScan SMPS (TSI, model 3910), and the Mini WRAS (Grimm, model 1371) were investigated against the WCCAP Mobility Particle Size Spectrometers (MPSS) and Condensation Particle Counters (CPC) using ambient aerosols and lab-generated PSL and NaCl particles. The inter-comparisons were performed both before and after the service and maintenance, and recommendations of timely service, maintenance and calibration were proposed, which will be instructive to the users. The manuscript is easy to follow and fits the scope of Atmospheric Measurement Techniques. However, I feel that the authors could provide more detailed work to demonstrate how service and maintenance improve the performance of the instruments, which will serve as valuable guidance for both existing and potential users. The reviewer recommends accepting this manuscript after addressing the following comments.

AR: Authors thank the reviewer for providing comments for improvement of the manuscript (MS).

Major comments:

RR:

1) The NanoScan SMPS and the Mini WRAS are portable and easy to use. But considering their inferiorities in both the time and size resolution, they may not be a great choice for mobile-platform measurement. Both the NanoScan SMPS and the Mini WRAS are not considered as fast (i.e., time resolution of 60 s). The new generation SMPS (e.g., TSI model 3938) can provide fast scan measurements (e.g., 15s and below, https://tsi.com/products/particle-sizers/scanning-mobility-particle-sizer-spectrometers/general-scanning-mobility-particle-sizer-(smps)-3938/).

AR: Authors agree with the reviewer that the TSI model 3938 can provide fast ultrafine particle measurements compared to NanoScan SMPS and Mini WRAS spectrometers.

TSI model 3938 is a more expensive research instrument and is less mobile, while portable instruments such as the NanoScan SMPS and the Mini WRAS are less advanced, however, can be well suited for mobile measurements (because of the light-weight) and especially point source identification. TSI model 3938 spectrometer is manufactured for different purposes, such as fast scans of particle number size distributions with relatively high particle number concentrations.

The purpose of this workshop was to identify the performance of the NanoScan SMPS and the Mini WRAS compared to high-end instruments such as the regular research mobility particle size spectrometers from different manufacturers or in this case to the reference mobility particle size spectrometer (MPSS) of the World Calibration Center.

RR:

2) If possible, I recommend the authors provide performance evaluations of the instruments based on their factory calibration, and analyze how the performance would change during long-term operations, such that the users could have a professional application note to follow.

AR:

Authors thank the reviewer for the suggestion. It is worth mentioning here that the factory calibration is somewhat different than the tests performed here. The units under test had a different history and different status of service and time since the last calibration. A defined use would be necessary to really be able to compare before and after a certain time of use. The "long-term operation" of the NanoScan SMPS is fine for e.g. daily use for several hours. However, the instrument is not designed for 24/7 (ambient) operation and therefore it is not advertised and should probably not be promoted with a performance test for this application. Isopropanol CPCs are rather suited for short-term measurements and not for continuous monitoring.

RR:

3) The author claimed that the TSI NanoScan SMPS instruments were significantly improved after service and maintenance, based on the comparisons of ambient PNSD measurement before and after maintenance. However, it is noticeable that the ambient PNSD of the two measurement periods are quite different. If the comparison is demonstrated by relative errors (i.e., concentration ratios), I suspect the relative error may still be comparable.

AR: Authors thank the reviewer for the valuable suggestion.

[Figure]

Figure 1: Relative deviation of the calculated total number concentration for the measured distribution relative to the average ambient aerosol distribution.

Here, in figure 1 (a) possible outliers have been identified. From figure 1(b), we suggest that maintenance of the instruments makes sense as all instruments are in the ±20% range.

RR:

4) By comparing Fig. 2 and Fig. 3, I am wondering why the TSI NanoScan SMPS failed to capture the size distribution of particles in the ultrafine mode from the PSL solution, but got reasonably good agreements when measuring ambient aerosols of the similar size range (i.e., 10 ~ 50 nm). Same for the GRIMM Mini WRAS (except for the one from UNICATT).

AR: TSI NanoScan SMPS and GRIMM Mini WRAS spectrometers cannot resolve the monodisperse peaks of single and doubly charged PSL particles due to the limited size resolution (cannot be changed by the user). The ambient particle number size distribution is broad and for PSL, it is quasi monodisperse. Therefore, almost all instruments failed to capture the PSL particle size distribution. If by chance a PSL size corresponding to the bin mid-point diameter would be chosen, the instrument would be able to resolve a monodisperse peak but that would be a special case.

RR:

5) If possible, please clarify how the sizing-relevant parameters (e.g., flow and/or voltage) look like before and after maintenance, especially the NanoScan SMPS from FMI2 which behaves quite differently after maintenance. I think that may help guide the users on how often to service the instrument.

AR:

The information related to flow has already been provided in supplementary tables before and after the maintenance period. However, there was no maintenance done on the voltage (HV of the DMA). For FMI2 instruments, the cyclone and charger are cleaned during servicing and maintenance day. In addition, wick and filters are changed.

RR:

6) With respect to the WCCAP MPSS, the NanoScan SMPS underestimated the PNC in the ultrafine aerosol mode for the intercomparison of ambient measurement, but overestimated the PNC when testing the polydisperse NaCl particles. Do the authors have any explanations for that?

AR: The NanoScan SMPS underestimated the PNC in the ultrafine aerosol mode when compared with WCCAP MPSS for ambient measurement. There is no obvious reason for this. We assume that the inversion matrix (unipolar charging) might be the reason for this discrepancy.

 Minor comments:

RR:

1) Table 1 row 1: Please also specify the particle counting technique of the NanoScan SMPS.

AR: This is corrected in the revised MS.

---

## Author Comment (AC2)

RR: Reviewer's response

AR: Author's response

Reviewer 2 comments:

RR:

The paper contains a set of measurements of several units of two models of portable self-contained nanoaerosol classifier-quantifier. The devices are evaluated against a reference instrument before and after servicing. Measurements against ambient aerosols and laboratory aerosols are performed in the size range of 10-200nm. Additional information regarding flow rate calibration, laser performance and time since last service is also included but not commented on. The comparison among various instruments of the same model is uncommon in the nano aerosol field and relevant for future publications, especially after long usage intervals. The reviewer recommends addressing the following comments prior to publication.

AR:

Authors thank the reviewer for providing comments for improvement of the manuscript (MS).

RR:

Annual calibration: "The results indicate that the portable instruments must be serviced and calibrated annually" The claim for this conclusion is not demonstrated. The flowmeter error and counting inaccuracy is measured for units with known last services, but (time-since-last-service vs counting-error[corrected for final inaccuracy]) is not displayed, therefore the statement "annually" has no basis.

AR:

The statement has been modified in the revised MS.

The revised statement is "The instruments need to be maintained and serviced according to the manufacturer's recommendations. For example, cleaning of NanoScan SMPS cyclones, zero check and flow measurement are tasks that have to be done regularly, also depending on the application (measurement time, concentration level etc.). The manufacturers recommend a professional annual service and calibration".

RR:

Time series ambient intercomparison: The methodology and usefulness for this measurement is unclear. I understand that the evaluated parameter is the stability over time of the overall counts. Additionally, since concentrations change over time, linearity can be evaluated Figures 9 a)b)c)d). The measurement method is confusing though. Particle number concentration over time incurs in the size dependent counting error. The Nanoscan SMPS ambient test, before and after the cleaning shows two clearly different particle distributions. A more relevant representation should evaluate the time dependent ambient aerosol per size channel or at a set of size channels, this way time linearity can be appropriately evaluated. Otherwise, if the ambient aerosol were to change its distribution counting will be affected. Figures 1b and 3b as plotted, suggest that you should not service your device as you will lose counting efficiency.

AR:

Authors thank the reviewer for this comment. The overall motive of using the time series ambient inter-comparison is to check the stability of counts over time.

[Figure]

(a)

[Figure]

(b)

Figure 1: Relative deviation of the calculated total number concentration for the measured distribution relative to the average ambient aerosol distribution.

The NanoScan SMPS ambient test, before and after the cleaning shows two clearly different particle distributions.  However, in figure 1 (a) possible outliers have been identified and after cleaning and maintenance all instruments were found to be in ±20% range (Figure 1 (b)). Figure 1 corresponds to the relative deviation of the calculated total number concentration for the measured distribution relative to the average ambient aerosol distribution.

So, from figure 1 (a) and 1(b), we can suggest that maintenance of the instruments makes sense even if there are two distinct particle distributions.

RR:

Nebulizer-generated NaCl aerosol: It is not specified the generation device/method for the precharged NaCl aerosol nor if the charger on the particle classifiers was turned off.

AR:

A custom-built nebulizer, similar to the TSI constant output atomizer, is used for the NaCl generation. The particles are dried using a diffusion drier and have due to the way of their generation a relatively high pre-charge. The bipolar charger in the classifier was used for the NaCl particles in the same way as for the ambient aerosol particles. Both portable instruments used their built-in unipolar charger. The charger in the particle classifier was not turned off.

RR:

There is little to no scientific value in including results from inaccurate or defective devices with regard to the GRIMM Mini WRAS spectrometers with SW <10 being uncalibrated. Unless a dedicated section addresses the background of said updates commenting on the inversion algorithm I would suggest removing any measurement data regarding said units. Data could be left as is, as a correction for previous publications with these instruments, if a comment is added.

AR:

Authors thank the reviewer for the suggestion. Actually, the direct correction of old data to the one recorded is not easily possible, since the inversion algorithms are in detail relatively different and require different correction factors. In addition, it cannot be guaranteed that the instrument would have behaved the same at an earlier time (would have had the same correction factor) as during the (later) calibration for the use of the newer software.

Therefore, we would like to keep the old data as it is, also showing the improvement in data inversion using the newer software version.

RR:

Regarding the test against 125nm PSL particles. This is quite a relevant plot, it provides a high concentration high resolution aerosol, which challenges both devices and shows the limitations of the instruments. For any of the cases it is not commented on the nature of the particles with aerodynamic diameters <<125nm. For 125nm the charging efficiencies of radioactive and corona should be similar but there is clearly a difference on the small size residues? A comment will help the reader with this plot. Maybe the devices did reach saturation

AR:

Authors thank the reviewer for considering the relevance of PSL tests carried out on portable instruments. The TSI NanoScan SMPS and GRIMM Mini WRAS spectrometers cannot resolve the monodisperse peaks of single and doubly charged PSL particles due to the limited size resolution. The ambient particle number size distribution is broad and for PSL it is quasi monodisperse. Therefore, these instruments are principally unable to capture a PSL number size distribution.